# Evolutionary Game Model of Internal Threats to Nuclear Security in Spent Fuel Reprocessing Plants Based on RDEU Theory

**Susu Ni** [1,*], **Shuliang Zou** [1,*] and **Jiahua Chen** [2]

1   School of Economics Management and Law, University of South China, Hengyang 421001, China
2   Hunan Provincial Key Laboratory of Emergency Safety Technology and Equipment for Nuclear Facilities, Hengyang 421001, China; chen.jiahua@usc.edu.cn
*   Correspondence: nisusu19970118@126.com (S.N.); zousl2013@126.com (S.Z.); Tel.: +86-155-8021-3051 (S.N.)

**Abstract:** The internal threat to nuclear security is one of the most serious problems in the physical protection supervision of spent fuel reprocessing plants. Both insiders and nuclear security departments have obvious characteristics of situational decision making and even irrational decision making. Combined with Game theory and RDEU theory, the RDEU Game model of insiders and nuclear security departments was constructed to analyze the existence of equilibrium solutions of two-way strategies under different emotional states. From a dynamic point of view, the influence and change process of emotion on participants' decision-making behavior were analyzed. Then, the model was numerically simulated to verify its accuracy and effectiveness, which showed that different emotional states and intensities would not only affect the final result of evolutionary equilibrium, but also change the evolution speed of the strategies. In addition, compared with insiders, the intensity of pessimism in the nuclear security department had a greater impact on the game equilibrium. Finally, we present some reasonable recommendations to prevent and protect nuclear security events at spent fuel reprocessing plants by strengthening the emotional supervision and guidance of insiders and the nuclear security department.

**Keywords:** nuclear security; insider threats; a spent fuel reprocessing plant; RDEU theory; emotional factors

## 1. Introduction

Internal personnel may be insiders and directly constitute internal threats. Insiders represent enemies who are authorized to enter nuclear facilities and physical protection systems, with the knowledge of the structure of nuclear facilities, dangerous radioactive sources, building environmental characteristics and other information [1]. Using the approved access rights and knowledge of nuclear facilities, insiders may betray the trust of the organization to bypass special nuclear security measures [2]. It is reported that the internal threats are much higher than external threats [3]. Insiders can make use of their understanding of the loopholes in the physical protection system to help external enemies carry out malicious acts such as illegal transfer, theft of nuclear materials or destruction of nuclear facilities [4]. Most known nuclear security events were carried out by insiders or at least with the help of insiders [5]. It is speculated that the theft of highly enriched uranium in Russia and the destruction of nuclear power plants in Belgium, which caused significant economic losses, were carried out by unidentified insiders [6]. Due to the high radioactivity in the spent fuel treatment process, as well as the closure of equipment, pipelines and radioactive solution valves in the heavy concrete equipment room or hot room, personnel cannot access nuclear materials. However, the radioactivity of nuclear products such as uranium dioxide and plutonium dioxide is very low; these are packaged in containers and accessible to personnel [7]. If nuclear materials and technology were acquired by insiders, it would have disastrous consequences [8,9]. Therefore, how to effectively deal with nuclear

security incidents from internal threats is an urgent problem to be solved in the physical protection supervision of spent fuel reprocessing plants.

Many scholars have tried to define the meaning of internal threat [10] and have put forward methods to prevent and mitigate internal threat nuclear security events from different angles, which can generally be summarized into two aspects. On the one hand, it is a management mechanism to limit the attempts of insiders to commit malicious acts [11,12]. The IAEA has issued preventive and protective measures against internal threats for qualitative analysis of potential internal threats to optimize personnel authority in key areas [13]. The US Nuclear Regulatory Commission proposed to expand the scope of internal insider background investigation [14]. On the other hand, it is the weakness analysis and effectiveness evaluation of the physical protection system to detect and prevent malicious acts committed by insiders. For example, Kim and others [15] and Zou Bowen and others [16] proposed a new method to evaluate the physical protection system by calculating the detection time and considering the characteristics of insiders. However, most of these studies focus on external constraints such as management systems and technical means and pay less attention to the impact of emotion on participants in internal threat nuclear security events [17]. Game theory can be used to analyze problems when two or more subjects make decisions that affect each other. Many scholars have introduced game theory into physical protection analysis. For example, Hefei and others [18] established a game model of complete rational attackers destroying virtual small reactors under complete information, considering that defenders are limited by budget. On this basis, Kim and others [15] modeled and analyzed the relative importance of various internal threat nuclear security events in the physical protection system and found that their severity depends on the authority and authorized access level of insiders. Rebecca and others [19] considered that attackers pay different attention to the quantity and quality of nuclear materials and proposed a method combining game theory and probability models to explore the best resource allocation strategy for the IAEA, so as to find illegal state behavior in the guaranteed Gas Centrifuge Enrichment Program. However, the previous literature assumes that participants are completely rational, and the irrational behavior of insiders has not been taken into account. The Rank-Dependent Expected Utility (RDEU) theory proposed by Quiggin [20] describes the impact of participants' emotions on game equilibrium results. Considering the irrational factors of participants, some scholars have applied RDEU theory to group conflict events, but most of them are between strong groups and vulnerable groups, such as land expropriation conflict (local government and farmers) [21], housing demolition compensation (local government and relocated people) [22] and environmental pollution avoidance (environmental avoidance enterprises and surrounding people) [23].

The prevention of and protection from nuclear security events is closely related to the organic combination of physical defense, technical defense and civil defense; since all systems are implemented and controlled by people, civil defense plays a decisive role in nuclear security [24]. However, both insiders and nuclear security departments have the characteristics of irrational decision making [25]. Insiders may take advantage of their understanding of equipment operation and access vouchers to carry out malicious acts on impulse due to dissatisfaction with the unit or society. The nuclear security department not only has an overly optimistic attitude towards the lack of vigilance of insiders, but also has an overwhelming anxiety about the malicious behavior of insiders [26]. First, a basic game model between insiders and nuclear security departments was established. Then, the RDEU Game model was constructed and solved. Finally, MATLAB software was used for numerical simulation.

## 2. RDEU Theory

According to the RDEU theory proposed by Quiggin [20], the following definitions are given.

**Definition 1.** *For random variable $X = \{x_i, i = 1, 2, \ldots, n\}$, the probability distribution of $X$ is $Pr\{X = x_i\} = P_i$, $i = 1, 2, \ldots, n$, where $p_i \geq 0$, $p_1 + p_2 + \ldots + p_n = 1$, each $x_i$ is sorted and $x_1 > x_2 > \ldots > x_n$ is specified, then the rank position $RP_i$ of $x_i$ is defined, as shown in Equation (1).*

$$RP_i = Pr\{X \leq x_i\} = p_i + p_{i+1} + \ldots + p_n, i = 1, 2, \ldots, n \tag{1}$$

Intuitively, the higher the rank position $RP_i$ of $x_i$, the greater the probability of not exceeding $x_i$, and the greater the weight of $x_i$ in decision making.

**Definition 2.** *The RDEU model means that the preference order "$\succ$" can be expressed by the real value function $V$ defined by the utility function $u$ and the emotion function $W$, that is, for random variables $X$ and $Y$, $X \succ Y \Leftrightarrow V(X, u, W) \succ V(Y, u, W)$. In a risk structure $\{p1, x1, \ldots, pn, xn\}$, the RDEU model is expressed as Equation (2):*

$$V(X, u, W) = \sum_{i=1}^{n} u(x_i) \cdot \pi(x_i) \tag{2}$$

Wherein, $\pi(x_i)$ is shown by Equation (3):

$$\pi(x_i) = W(p_1 + p_2 + \ldots + p_i) - W(p_1 + p_2 + \ldots + p_i - 1), i = 1, 2, \ldots, n \tag{3}$$

According to Equations (1) and (2), Equation (4) can be obtained:

$$\pi(x_i) = W(p_i + 1 - RP_i) - W(1 - RP_i), i = 1, 2, \ldots, n \tag{4}$$

$\pi(x_i)$ is the function determining the weight, where $\pi(x1) = W(p1)$, and the emotion function $W(\cdot)$ is a monotonic increasing function, satisfying $W(0) = 0$ and $W(1) = 1$. If and only if $W$ is a convex function, $\pi(x_i)$ is monotonically decreasing with respect to the rank position $RP_i$. Furthermore, if and only if $W$ is a concave function, the level $\pi(x_i)$ is monotonically increasing with respect to the rank position $RP_i$.

## 3. RDEU Game Model of Internal Threat Nuclear Security Events

### 3.1. Problem Description

In the physical protection supervision of nuclear materials and nuclear facilities in fixed places such as spent fuel reprocessing plants, the operating unit of spent fuel reprocessing plants is responsible for formulating, revising and organizing the implementation of various rules and regulations. The nuclear security department is jointly composed of armed police, security guards and security personnel to detect, suspend and respond to the malicious acts of insiders by implementing preventive and protective measures.

Insiders A and the nuclear security department B are taken as the research objects. Insiders have two choices: implementing malicious behaviors and not implementing malicious behaviors. The implementation of malicious behavior refers to the theft, illegal transfer and destruction of nuclear materials and nuclear facilities.

The nuclear security department has two strategic choices, which are strict implementation and non-strict implementation. The implementation strength $\sigma$ of the nuclear security department for the malicious acts of insiders meets $\sigma = 1$, which includes the strict implementation of various rules and regulations, personal protection provisions, the most powerful testing means and the most stringent punishment system. If it is not strictly implemented, the implementation strength $\sigma$ meets $0 < \sigma < 1$.

### 3.2. Assumptions

**Assumption 1.** *Referring to the research of Zou Bowen and others [27], when analyzing the malicious behavior of insiders, it is generally considered that the insiders are non-violent and the purpose of the crime is to steal uranium dioxide and plutonium dioxide, regardless of radioactive damage.*

**Assumption 2.** *Both insiders and the nuclear security department aim at maximizing their own interests. When both sides do not have complete information, the probability of the insiders choosing the two strategies of "implementing malicious behaviors" and "not implementing malicious behaviors" is m and $1 - m$ ($0 < m < 1$). The probability of nuclear security department choosing "strict implementation" and "non-strict implementation" strategies is n and $1 < n$ ($0 < n < 1$).*

**Assumption 3.** *Based on the research of Yang Guangming and Shi Yanjun [28], the reward and punishment mechanism is introduced. It is assumed that the nuclear security department strictly implements the prevention and protection measures. When insiders commit malicious acts, the punishment for them is F and the reward for the nuclear security department is K (reward from the operating unit of the spent fuel reprocessing plant). Assuming that the nuclear security department does not strictly implement the prevention and protection measures, when insiders commit malicious acts, the punishment for them is $\sigma F$ and the punishment for the nuclear security department is T (punishment from the operating unit of the spent fuel reprocessing plant).*

**Assumption 4.** *Assuming that the nuclear security department strictly implements prevention and protection measures, it is certain to detect the malicious behavior of insiders, that is, $P_{ND1} = 0$.*

**Assumption 5.** *Participants have three emotional states: pessimism, optimism and rationality. According to the RDEU theory, the emotional function of insiders is $W_A(m) = m^{r1}$ ($r_1$ is the emotional index of insiders, and $r_1 > 0$) and the emotional function of the nuclear security department is $W_B(n) = n^{r2}$ ($r_2$ is the emotional index of nuclear security department, and $r_2 > 0$).*

*3.3. Basic Game Model between Insiders and Nuclear Security Department*

In order to enhance the readability of the article, the significance of the parameters involved in the evolutionary game model is described below to provide a basis for the later construction of the RDEU model of internal threat nuclear security events. The symbols and significance of relevant parameters are shown in Table 1. The value range of the parameters involved is non negative, and generally, $F > I_A - C_1$, $P_{ND2}S > C_1 + \sigma F$, $K + F > C_2 > C_3$, $T > I_B + \sigma F$.

**Table 1.** Definition of model parameters.

| Parameters | Definition of Parameter Symbols |
|---|---|
| $m$ | The proportion of insiders who choose to implement malicious behavior, $0 < m < 1$. |
| $n$ | The proportion of the nuclear security department choosing to strictly implement the strategy, $0 < n < 1$. |
| $I_A$ | Stable basic income obtained by insiders. |
| $I_B$ | Stable basic benefits obtained by the nuclear security sector. |
| $C_1$ | The cost of malicious acts committed by insiders. |
| $C_2$ | The cost of economic and human security inspection paid by the nuclear security department to strictly implement preventive and protective measures. |
| $C_3$ | The economic and human security costs incurred by the nuclear security department for not strictly implementing preventive and protective measures, $C_2 > C_3$. |
| $\sigma$ | Enforcement of malicious acts of insiders by nuclear security department (detection means and reward and punishment system). |
| $S$ | When the nuclear security department does not strictly implement preventive and protective measures, the insiders can obtain the economic benefits and psychological benefits of products such as uranium dioxide and plutonium dioxide from malicious acts. |
| $P_{ND1}$ | When the nuclear security department strictly implements physical protection measures, the probability that malicious acts committed by insiders will not be detected, $0 \leq P_{ND1} \leq 1$. |
| $P_{ND2}$ | When the nuclear security department does not strictly implement physical protection measures, the probability that malicious acts committed by insiders will not be detected, $0 \leq P_{ND1} \leq P_{ND2} \leq 1$. |
| $F$ | The nuclear security department chose to strictly implement the strict punishment for malicious acts committed by insiders. |
| $T$ | If the nuclear security department fails to strictly implement the preventive and protective measures after the malicious acts of the insiders, it will be punished by the operating unit of the spent fuel reprocessing plant. |
| $K$ | The reward from the operating unit of the spent fuel reprocessing plant that the nuclear security department can obtain by strictly implementing the prevention and protection measures after the malicious acts of the insiders. |

According to the above description, the game income matrix between the internal insider and the nuclear security department can be obtained, as shown in Table 2, and Equations (5) and (6) are valid.

$$I_A + P_{ND1}S - C_1 - \sigma F > I_A = I_A > I_A + P_{ND1}S - C_1 - F \tag{5}$$

$$I_B - C_2 + K + F > I_B - C_3 > I_B - C_2 > I_B - C_3 - T + \sigma F \tag{6}$$

**Table 2.** Game income matrix between insiders and nuclear security department.

| Nuclear Security Department (B) | Insiders (A) | |
|---|---|---|
| | Implementing Malicious Behaviors (*m*) | Not Implementing Malicious Behaviors (1−*m*) |
| Strict Implementation (*n*) | $I_B - C_2 + K + F, I_A + P_{ND1}S - C_1 - F$ | $I_B - C_2, I_A$ |
| Non-Strict Implementation (1 − *n*) | $I_B - C_3 - T + \sigma F, I_A + P_{ND2}S - C_1 - \sigma F$ | $I_B - C_3, I_A$ |

During the prevention and protection of internal threat nuclear security events, due to the sudden nature of the situation, both insiders and the nuclear security department may have strong pessimism or optimism. In order to further analyze the influence of emotional factors on the equilibrium solution, the RDEU theory is used to expand the form of utility function of both sides. Different emotional factors will cause a certain deviation to the participants' subjective probability of event occurrence. Pessimism will reduce the probability of events, while optimism is the opposite. Therefore, the subjective probability function affected by emotional factors is expressed as $W(p_i) = p_i{}^r$, $i = 1, 2, \ldots, n$, where $r$ is the emotional index of participant $i$, and $p_i$ is the objective probability of event $x_i$ and $p_i \in [0, 1]$. According to the value range of objective probability, when $r_i = 1$, the players do not have emotion, that is, they are in a rational state. When $r_i > 1$, they subjectively underestimate the occurrence probability and call it "pessimistic". According to Equation (3), the weight of the event in the utility function will rise. When $0 < r_i < 1$, the probability of occurrence is subjectively overestimated, which is called "optimism". Similarly, the weight of this event in the utility function will be reduced. Specifically, as shown in Figure 1, the vertical axis in the figure represents the probability cumulative distribution, that is, the emotion function $W$. The concave curve indicates that the probability of events will be underestimated under pessimism. For event $x_i$, the weight $\pi(x_i)$ in the utility function under pessimism will be greater than the weight $\Delta p_i$ under rational situations (without emotion).

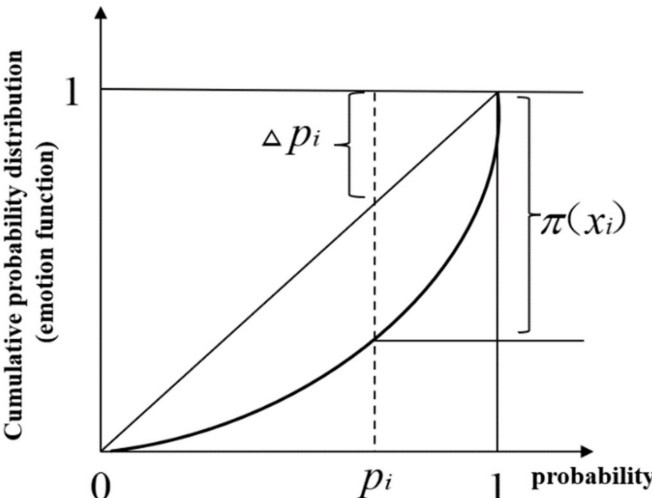

**Figure 1.** Schematic diagram of the weight of event $x_i$ in the utility function under pessimism.

In order to facilitate the analysis, the income of both insiders and the nuclear security department is simplified and sorted, so that, $a = I_B - C_2 + K + F$, $b = I_B - C_2$, $c = I_B - C_3 - T + \sigma F$, $d = I_B - C_3$, $e = I_A + P_{ND1}S - C_1 - F$, $f = I_A = h$, $g = I_A + P_{ND2}S - C_1 - \sigma F$. Then, there

are inequalities $a > d > b > c$, $g > f = h > e$; the income improvement matrix is shown in Table 3.

**Table 3.** Game income improvement matrix between insiders and nuclear security department.

| Nuclear Security Department (B) | Insiders (A) | |
| --- | --- | --- |
| | Implementing Malicious Behaviors (*m*) | Not Implementing Malicious Behaviors (1−*m*) |
| Strict Implementation (*n*) | *a, e* | *b, f* |
| Non-Strict Implementation (1 − *n*) | *c, g* | *d, h* |

### 3.4. RDEU Model between Insiders and Nuclear Security Department

Based on the above assumptions, the game income improvement matrix between the insiders and the nuclear security department was established. According to the RDEU theory and the above analysis, the strategic income, probability, rank position and decision-making weight of insiders and the nuclear security department can be obtained, as shown in Tables 4 and 5 below. Specifically, when the strategic benefit of insiders is $e$, the probability of their malicious behavior is $mn$ (see Table 3). From $g > f = h > e$ and Equation (1), it can be seen that its value of rank position is equal to the value of probability, i.e., $mn$.

**Table 4.** $p_i$, $RP_i$ and $\pi(x_i)$ of each strategy of insiders.

| Strategic Benefits of Insiders $u(x_i)$ | Probability $p_i$ | Rank Rotation $RP_i$ | Decision Weight $\pi(x_i)$ |
| --- | --- | --- | --- |
| $g$ | $m(1-n)$ | $1$ | $W_A(m - mn)$ |
| $f = h$ | $1 - m$ | $1 - m + mn$ | $W_A(1 - mn) - W_A(m - mn)$ |
| $e$ | $mn$ | $mn$ | $1 - W_A(1 - mn)$ |

**Table 5.** $p_i$, $RP_i$ and $\pi(x_i)$ of each strategy of nuclear security department.

| Strategic Benefits of Nuclear Security Department $u(x_i)$ | Probability $pi$ | Rank Position $RP_i$ | Decision Weight $\pi(x_i)$ |
| --- | --- | --- | --- |
| $a$ | $mn$ | $1$ | $W_B(mn)$ |
| $d$ | $(1-m)(1-n)$ | $1 - mn$ | $W_B(1 - m - n + 2mn) - W_B(mn)$ |
| $b$ | $n(1-m)$ | $m + n - 2mn$ | $W_B(1 - m + mn) - W_B(1 - m - n + 2mn)$ |
| $c$ | $m(1-n)$ | $m(1-n)$ | $1 - W_B(1 - m + mn)$ |

The expected utility of the insider adopting the strategy of implementing malicious behaviors is assumed to be $U_{A1}$, as shown in Equation (7). According to Equation (2), the expected utility function of the insiders' RDEU model is $V_A(X, u, W)$, as shown in Equation (8).

$$U_{A1} = e \cdot W_B(n) + g \cdot [1 - W_B(n)] = e \cdot n^{r2} + g \cdot (1 - n^{r2}) = g + (e - g) \cdot n^{r2} \quad (7)$$

$$
\begin{aligned}
V_A(X, u, W) &= \sum_{i=1}^{n} u(x_i) \cdot \pi(x_i) = g \cdot W_A(m - mn) + f \cdot [W_A(1 - mn) - W_A(m - mn)] \\
&+ e \cdot [1 - W_A(1 - mn)] \\
&= (g - f) \cdot (m - mn)^{r1} + (f - e) \cdot (1 - mn)^{r1} + e
\end{aligned}
\quad (8)
$$

Similarly, it is recorded that the expected utility of the security department adopting the strict implementation strategy is $U_{B1}$, as shown in Equation (9). The expected utility function of the RDEU model of the nuclear security department is $V_B(X, u, W)$, as shown in Equation (10).

$$U_{B1} = a \cdot W_A(m) + b \cdot [1 - W_A(m)] = a \cdot m^{r1} + b \cdot (1 - m^{r1}) = b + (a - b) \cdot m^{r1} \quad (9)$$

$$
\begin{aligned}
V_B(X, u, W) &= \sum_{i=1}^{n} u(x_i) \cdot \pi(x_i) = a \cdot W_B(mn) + d \cdot [W_B(1 - m - n + 2mn) - W_B(mn)] + \\
&\quad b \cdot [W_B(1 - m + mn) - W_B(1 - m - n + 2mn)] + c \cdot [1 - W_B(1 - m + mn)] \\
&= (a - d) \cdot (mn)^{r2} + (d - b) \cdot (1 - m - n + 2mn)^{r2} + (b - c) \cdot (1 - m + mn)^{r2} + c
\end{aligned}
\tag{10}
$$

In the game of internal threat nuclear security events, the behavior strategy adjustment process between insiders and the nuclear security department can be described by replication dynamic equations, as shown in Equations (11) and (12).

$$
\frac{dm}{dt} = m^{r1} \cdot (U_{A1} - EU_A) = m^{r1} \cdot [(g - e) \cdot (1 - n^{r2}) - (g - f) \cdot (m - mn)^{r1} - (f - e) \cdot (1 - mn)^{r1}]
\tag{11}
$$

$$
\frac{dn}{dt} = n^{r2} \cdot (U_{B1} - EU_B) = n^{r2} \cdot \{(b - c) \cdot [1 - (1 - m + mn)^{r2}] + (a - b) \cdot m^{r1} - (a - d) \cdot (mn)^{r2} - \\
(d - b) \cdot (1 - m - n + 2mn)^{r2}\}
\tag{12}
$$

From the above two equations, five evolutionary equilibrium points can be obtained: $E_1(0,0)$, $E_2(0,1)$, $E_3(1,0)$, $E_4(1,1)$, and $E_5(n^*, m^*)$. The value of the fifth equilibrium point can be solved by the transcendental Equation (13).

$$
\begin{cases}
(g - e) \cdot (1 - n^{r2}) - (g - f) \cdot (m - mn)^{r1} - (f - e) \cdot (1 - mn)^{r1} = 0 \\
(b - c) \cdot [1 - (1 - m + mn)^{r2}] + (a - b) \cdot m^{r1} - (a - d) \cdot (mn)^{r2} - (d - b) \cdot (1 - m - n + 2mn)^{r2} = 0
\end{cases}
\tag{13}
$$

### 3.5. Stability Analysis of Game Model

According to the evolutionary equilibrium theory, the stability of the five equilibrium points can be judged according to the condition that the Jacobian matrix satisfies $\boldsymbol{Det(J)} > 0$ and $\boldsymbol{Tr(J)} < 0$, and then the evolutionary stability strategy under the influence of emotional factors is discussed. Therefore, the Jacobian matrix of the RDEU model between the above insiders and the nuclear security department is: $\boldsymbol{J} = \begin{bmatrix} \frac{\partial F(n)}{\partial n} & \frac{\partial F(n)}{\partial m} \\ \frac{\partial F(m)}{\partial n} & \frac{\partial F(m)}{\partial m} \end{bmatrix}$ where $F(n) = \frac{dn}{dt}$, $F(m) = \frac{dm}{dt}$. In combining Equations (11) and (12), values of $\frac{\partial F(n)}{\partial n}$, $\frac{\partial F(n)}{\partial m}$, $\frac{\partial F(m)}{\partial n}$ and $\frac{\partial F(m)}{\partial m}$ can be obtained as shown in Equations (14)–(17).

$$
\begin{aligned}
\frac{\partial F(n)}{\partial n} &= n^{r2} \cdot [m \cdot r2 \cdot (c - b) \cdot (mn - m + 1)^{r2-1} - m \cdot r2 \cdot (mn)^{r2-1} \cdot (a - d) + r2 \cdot (2m - 1) \cdot (b - d) \cdot \\
&\quad (2mn - n - m + 1)^{r2-1}] + n^{r2-1} \cdot r2 \cdot \{m^{r1} \cdot (a - b) + (c - b) \cdot [(m \cdot n - m + 1)^{r2} - 1] - (mn)^{r2} \cdot \\
&\quad (a - d) + (b - d) \cdot (2mn - n - m + 1)^{r2}\}
\end{aligned}
\tag{14}
$$

$$
\begin{aligned}
\frac{\partial F(n)}{\partial m} &= n^{r2} \cdot [m^{r1-1} \cdot r1 \cdot (a - b) - n \cdot r2 \cdot (mn)^{r2-1} \cdot (a - d) + r2 \cdot (c - b) \cdot (n - 1) \cdot (mn - m + 1)^{r2-1} + \\
&\quad r2 \cdot (2n - 1) \cdot (b - d) \cdot (2mn - n - m + 1)^{r2-1}]
\end{aligned}
\tag{15}
$$

$$
\frac{\partial F(m)}{\partial n} = m^{r1} \cdot [n^{r2-1} \cdot r2 \cdot (e - g) + m \cdot r1 \cdot (m - m \cdot n)^{r1-1} \cdot (g - f) - m \cdot r1 \cdot (e - f) \cdot (1 - mn)^{r1-1}]
\tag{16}
$$

$$
\begin{aligned}
\frac{\partial F(m)}{\partial m} &= m^{r1} \cdot [r_1 \cdot (m - mn)^{r1-1} \cdot (g - f) \cdot (n - 1) - n \cdot r1 \cdot (e - f) \cdot (1 - mn)^{r1-1}] + m^{r1-1} \cdot r1 \cdot [(e - \\
&\quad f) \cdot (1 - mn)^{r1} + (e - g) \cdot (n^{r2} - 1) - (n - mn)^{r1} \cdot (g - f)]
\end{aligned}
\tag{17}
$$

The emotions of insiders and the nuclear security department are divided into three dimensions: pessimism ($r > 1$), optimism ($r < 1$) and rational state ($r = 1$). The value of emotion index $r$ represents the intensity of emotion. The stability of the equilibrium point when the two sides of the game are in different emotional states is discussed below.

#### 3.5.1. Both Players in the RDEU Model Were Rational

MATLAB software was used for calculation, and $(n^*, m^*) = (\frac{g-f}{g-e}, \frac{d-b}{a-c+d-b})$ could be obtained. Therefore, when both sides of the game were in a rational state without emotion, the insiders chose to implement malicious behavior with a probability of $\frac{d-b}{a-c+d-b}$, and the nuclear security department chose to strictly implement preventive and protective

measures with a probability of $\frac{g-f}{g-e}$, that is, the equilibrium point of the hybrid strategy was $E_5(n^*, m^*)$. See Table 6 for the judgment of the stability of corresponding equilibrium points.

**Table 6.** Evolution state judgment table when participants were in a rational state.

| Equilibrium Point | $\frac{\partial F(n)}{\partial n}$ | $\frac{\partial F(n)}{\partial m}$ | $\frac{\partial F(m)}{\partial n}$ | $\frac{\partial F(m)}{\partial m}$ | $Det(J)$ | $Tr(J)$ | Local Stability |
|---|---|---|---|---|---|---|---|
| $E_1(0,0)$ | $b-d$ | $0$ | $0$ | $g-f$ | $-$ | Uncertainty | Instability |
| $E_2(0,1)$ | $d-b$ | $0$ | $0$ | $e-f$ | $-$ | Uncertainty | Instability |
| $E_3(1,0)$ | $a-c$ | $0$ | $0$ | $f-g$ | $-$ | Uncertainty | Instability |
| $E_4(1,1)$ | $c-a$ | $0$ | $0$ | $f-e$ | $-$ | Uncertainty | Instability |
| $E_5(n^*,m^*)$ | $0$ | $(a-c+d-b)\cdot(n-n^2)$ | $(g-e)\cdot(m^2-m)$ | $0$ | $+$ | $0$ | Saddle point |

As shown in Table 6, when both sides of the game were in a rational state without emotion, equilibrium points $E_1(0,0)$, $E_2(0,1)$, $E_3(1,0)$ and $E_4(1,1)$ met $Det(J) < 0$, and the value of $Tr(J)$ was uncertain, not reaching a stable state. The equilibrium point $E_5(n^*, m^*)$ of the hybrid strategy was $Det(J) > 0$ and $Tr(J) = 0$, so it was neither asymptotically stable nor unstable. Therefore, if participants are in a rational state, they will constantly adjust their strategies according to the situation. However, the assumption of a rational state without emotion is difficult to replicate in reality.

### 3.5.2. The Nuclear Security Department Was Rational and Insiders Were Emotional

When the choice of the nuclear security department was not affected by emotional factors, and the choice of insiders was affected by their own pessimism and optimism, $r_1 \neq 1$ and $r_2 = 1$. This combination of emotional states is the most common in real internal threat nuclear security events. See Table 7 for the judgment of the stability of corresponding equilibrium points.

**Table 7.** The nuclear security department was rational and insiders were emotional.

| Equilibrium Point | $\frac{\partial F(n)}{\partial n}$ | $\frac{\partial F(n)}{\partial m}$ | $\frac{\partial F(m)}{\partial n}$ | $\frac{\partial F(m)}{\partial m}$ | $Det(J)$ | $Tr(J)$ | Local Stability |
|---|---|---|---|---|---|---|---|
| $E_1(0,0)$ | $b-d$ | $0$ | $0$ | $0$ | $0$ | $-$ | Instability |
| $E_2(0,1)$ | $d-b$ | $b-a$ | $0$ | $0$ | $0$ | $+$ | Instability |
| $E_3(1,0)$ | $a-c$ | $0$ | $(r_1-1)\cdot(g-e)$ | $r_1\cdot(f-g)$ | $-$ | Uncertainty | Instability |
| $E_4(1,1)$ | $c-a$ | $(r_1-1)\cdot(a-b)$ | $e-g$ | $0$ | $+/-$ | $-$ | Stability/Instability |
| $E_5(n^*,m^*)$ | It depends on the return value of both parties and the emotional index $r_1$ of internal insiders | | | | | | |

As shown in Table 7, $Det(J)$ and $Tr(J)$ corresponding to $E_1(0,0)$, $E_2(0,1)$ and $E_3(1,0)$ did not meet the evolutionary stability conditions. For $E_4(1,1)$, when insiders were pessimistic ($r1 > 1$). With the condition of $Det(J) > 0$ and $Tr(J) < 0$, $E_4(1,1)$ was the stable point. When insiders were optimistic ($0 < r1 < 1$), the values of $Det(J)$ and $Tr(J)$ of $E_4(1,1)$ were negative, with the equilibrium point being unstable.

This result showed that pessimistic insiders tended to implement malicious behaviors, while the rational nuclear security department tended to strictly implement preventive and protective measures. Table 7 also shows that it was difficult to judge the stability of $E_5(n^*, m^*)$. When the income values of both sides of the game and the emotional index $r_1$ of insiders are different, different results will be produced, and the equilibrium point $E_5(n^*, m^*)$ may be a stable solution. See more analysis in the numerical simulation section.

### 3.5.3. Insiders Were Rational and the Nuclear Security Department Was Emotional

At this time, insiders remained calm, i.e., $r_1 = 1$. However, the nuclear security department had overly optimistic attitudes, such as blind trust in insiders, or pessimistic emotions, such as anxiety and panic that they could not be prevented from committing crimes against internal enemies, that is, $r_2 \neq 1$. See Table 8 for the judgment of the stability of corresponding equilibrium points.

**Table 8.** Insiders were rational and the nuclear security department was emotional.

| Equilibrium Point | $\frac{\partial F(n)}{\partial n}$ | $\frac{\partial F(n)}{\partial m}$ | $\frac{\partial F(m)}{\partial n}$ | $\frac{\partial F(m)}{\partial m}$ | $Det(J)$ | $Tr(J)$ | Local Stability |
|---|---|---|---|---|---|---|---|
| $E_1(0,0)$ | 0 | 0 | 0 | $g-f$ | 0 | $+$ | Instability |
| $E_2(0,1)$ | 0 | $a-b$ | 0 | $e-f$ | 0 | $-$ | Instability |
| $E_3(1,0)$ | 0 | 0 | $g-e$ | $f-g$ | 0 | $-$ | Instability |
| $E_4(1,1)$ | $r_2 \cdot (c-a)$ | $(1-r_2) \cdot (a-b)$ | $(1-r_2) \cdot (g-e)$ | $f-e$ | $-$ | Uncertainty | Instability |
| $E_5(n^*,m^*)$ | It depends on the income value of both parties and the sentiment index $r_2$ of the nuclear security department | | | | | | |

As shown in Table 8, $Det(J)$ and $Tr(J)$ corresponding to equilibrium points $E_1(0,0)$, $E_2(0,1)$, $E_3(1,0)$ and $E_4(1,1)$ did not meet the evolutionary stability conditions and were not stable points. In addition, insiders were not emotional, and the nuclear security department did not tend to be purely strategic when it was pessimistic or optimistic. For the fifth point $E_5(n^*,m^*)$, the equilibrium stability result is related to the emotion index $r_2$ of the nuclear security department and the income value of both parties in the game of internal threat nuclear security events. Different parameter values will lead to different game results.

3.5.4. Insiders and the Nuclear Security Department Were Emotional

In the event of an internal threat to nuclear security, both insiders and the nuclear security department may be emotional, resulting in optimistic or pessimistic cognitive bias when making strategic choices, i.e., $r_1 \neq 1$, $r2 \neq 1$. At this time, if $n = n^*$ and $m = m^*$ to make Equation (13) true, it showed that the game model had a mixed strategy Nash equilibrium solution under the influence of emotional factors and there was no Nash equilibrium solution under the condition of extreme optimism $(r_1, r_2 \to 0)$ or extreme pessimism $(r_1, r_2 \to +\infty)$.

As shown in Table 9, equilibrium points $E_1(0,0)$, $E_2(0,1)$ and $E_3(1,0)$ did not meet the evolutionary stability conditions, but $Tr(J)$ of $E_1(0,0)$ and $E_2(0,1)$ was zero, which belonged to the saddle point. This means that for the equilibrium point $E_5(n^*,m^*)$ of the hybrid strategy, the hybrid strategy may evolve into a stable strategy under the influence of the emotions of insiders and the nuclear security department. For equilibrium point $E_4(1,1)$, if both sides of the game were pessimistic (the stronger the pessimism, the greater the value of $r_2$), the sentiment of the nuclear security department was stronger. Or, when both sides were optimistic (the stronger the optimism, the smaller the value of $r_1$), insiders' emotions were stronger. In addition to the optimism held by insiders and the pessimism held by the nuclear security department, there were evolutionary stability strategies. That is, no matter what emotional combination the two sides hold, as long as $r_1 < r_2$ is satisfied, the game system has an evolutionary stable solution.

**Table 9.** Insiders and the nuclear security department were emotional.

| Equilibrium Point | $\frac{\partial F(n)}{\partial n}$ | $\frac{\partial F(n)}{\partial m}$ | $\frac{\partial F(m)}{\partial n}$ | $\frac{\partial F(m)}{\partial m}$ | $Det(J)$ | $Tr(J)$ | Local Stability |
|---|---|---|---|---|---|---|---|
| $E_1(0,0)$ | 0 | 0 | 0 | 0 | 0 | 0 | saddle point |
| $E_2(0,1)$ | 0 | 0 | 0 | 0 | 0 | 0 | saddle point |
| $E_3(1,0)$ | 0 | 0 | $r_1 \cdot (g-e)$ | $r_1 \cdot (f-g)$ | 0 | $-$ | Instability |
| $E_4(1,1)$ | $r_2 \cdot (c-a)$ | $(r_1-r_2) \cdot (a-b)$ | $r_2 \cdot (e-g)$ | 0 | $+/-$ | $-$ | Stability/Instability |
| $E_5(n^*,m^*)$ | In the case of extreme optimism $(r_1, r_2 \to 0)$ or extreme pessimism $(r_1, r_2 \to +\infty)$, there is no Nash equilibrium solution | | | | | | |

From the above analysis, the result that "participant sentiment will affect the evolutionary equilibrium and stability of internal threat nuclear security events" can be drawn. Furthermore, compared with the rational state, different emotion combinations have different effects on the equilibrium state of evolutionary game. Emotion can promote the transformation from pure strategy to evolutionary stability strategy. Moreover, if both parties involved in the game choose the mixed strategy, for Nash equilibrium point $E_5(n^*,m^*)$, the emotion of insiders and nuclear security department will affect its probability and

stability, and the mixed strategy may transition from pure strategy to evolutionary stability strategy.

## 4. Numerical Simulation Analysis

The above analysis showed that the emotional state of the game participants had an important impact on the evolution of internal threat nuclear security events. MATLAB software was selected to draw the simulation diagram of the impact of different emotional states and emotional intensity combinations of insiders and the nuclear security department, so as to investigate the practicability of the model. Since it was difficult to obtain the relevant data of such nuclear security events, under the condition of meeting the income relationship described above, the specific numerical assumptions of the parameters of the game-improved income matrix were $a = 1$, $b = -3$, $c = -4$, $d = 0$, $e = -3$, $f = h = 0.5$, $g = 3$ and were carried out in the initial state of $m = n = 0.5$ [14,18].

### 4.1. All Participants Were in a Rational State

As shown in Figure 2, when $r_1 = r_2 = 1$, the evolution path rises and falls one after another, and insiders and the nuclear security department have not formed an evolutionary stability strategy. This shows that when both parties were in a rational state, they would implement malicious behaviors and strictly implement preventive and protective measures with a certain probability. Because insiders had legal authority, they may not need to continuously carry out malicious acts, which made these difficult to be found in the short term, as they may turn into a long-term latent state under the deterrence strictly implemented by the nuclear security department. In addition, the nuclear security department would not easily believe that insiders would betray the trust of the organization, and it would pay a high cost to impose a state of total martial law. Therefore, the nuclear security department would selectively and strictly implement preventive and protective measures with a certain probability. At this time, participants of internal threat nuclear security events were facing a long-term "cat and mouse" trend, and could not form a consistent strategy.

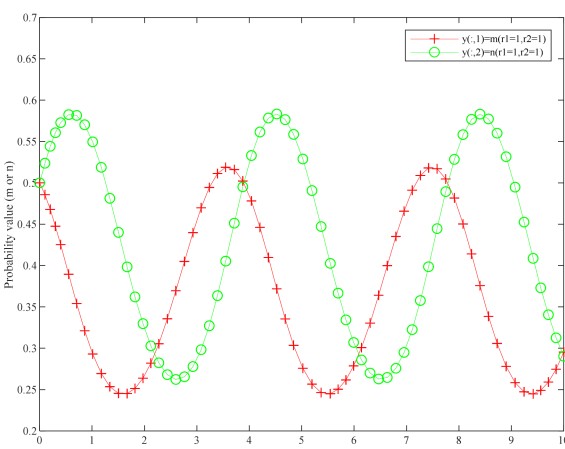

**Figure 2.** Decision-making simulation of rational participants.

### 4.2. The Influence of Insiders' Emotion on Evolutionary Equilibrium

As shown in Figure 3, when $r_1 = 1.5$ and $r_2 = 1$, the probability of malicious behavior by slightly pessimistic insiders and the probability of strict implementation strategy by the rational nuclear security department move rapidly close to one. At this time, there was an evolutionary stability strategy (strict implementation and implementing malicious behaviors). Increasing the pessimistic intensity of insiders to two showed the same result, but the evolution process of the game was significantly shortened, and the strategies of both groups converged to a stable state at a faster speed. In this case, insiders used their understanding of equipment operation, access vouchers and other emotional behaviors due to radical emotions such as panic and anger towards the unit or society. At the same

time, these emotions provided a kind of negative cognition for insiders, leading to the continuous strengthening of emotions and the making of irrational choices. The rational nuclear security department quickly judged the situation. In the face of the rising negative psychological situation of insiders, in order to avoid a lasting chain reaction, it had to take immediate measures to strictly implement prevention and protection measures and adopt the strictest detection means and the strictest punishment system for the malicious acts of insiders, so as to form a kind of stabilization strategy (implementing malicious behaviors and strict implementation). Therefore, the nuclear security department needs to strengthen the supervision of the psychological state of insiders, accurately judge the emotional state of insiders, and then decide whether to strictly implement preventive and protective measures at all times.

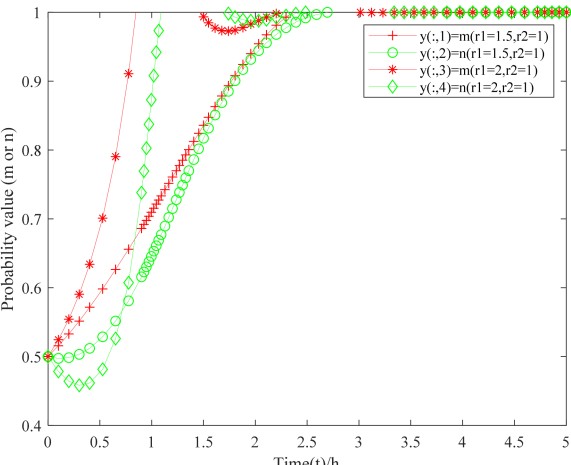

**Figure 3.** Decision-making simulation of pessimistic insiders.

As shown in Figure 4, when $r_1 = 0.8, r_2 = 1$, the probability of slightly optimistic insiders committing malicious acts and the probability of the rational nuclear security department adopting strict implementation strategies are about 0.31 and 0.34. Although no stabilization strategy has been formed, if the optimism intensity of insiders is increased to 0.5, the probability of malicious behavior by insiders and the probability of strict implementation by the nuclear security department will be more quickly stabilized at about 0.21 and 0.27. That is, in the rational state of the nuclear security department, the greater the intensity of optimism of insiders (the smaller the emotional index $r_1$), the more conducive it is to form a better result (non-strict implementation and not implementing malicious behaviors). This means that with the increase of the intensity of insiders' optimism, insiders will be more inclined not to implement malicious behavior strategies. When the nuclear security department is aware of this in a rational state, it will give more trust to insiders and then adopt a lax implementation strategy.

In addition, compared with optimism, pessimism such as disappointment and anxiety is more likely to drive insiders to collude and then take concerted action to accelerate the occurrence of internal threats to nuclear security.

### 4.3. The Impact of Nuclear Security Department Sentiment on Evolutionary Equilibrium

As shown in Figure 5, when $r_1 = 1, r_2 = 1.5$, that is, the nuclear security department is pessimistic and the internal insider remains rational. The strict implementation probability of the nuclear security department quickly approaches one, while the probability of malicious behavior by insiders decreases to zero after a period of improvement. At this time, there is a stable strategy (strict implementation and not implementing malicious behaviors). Keeping the rational state of insiders unchanged and increasing the pessimistic intensity of the nuclear security department to two shows similar results, and the strategies of both sides of the game converge to a stable state faster. This shows that under the pessimistic

state, the nuclear security department will be vigilant and doubt the insiders, and the rational insiders are aware of this. Although they tend to implement the malicious behavior strategy in the initial stage, they will eventually adopt the strategy of not implementing the malicious behavior in order to avoid losses.

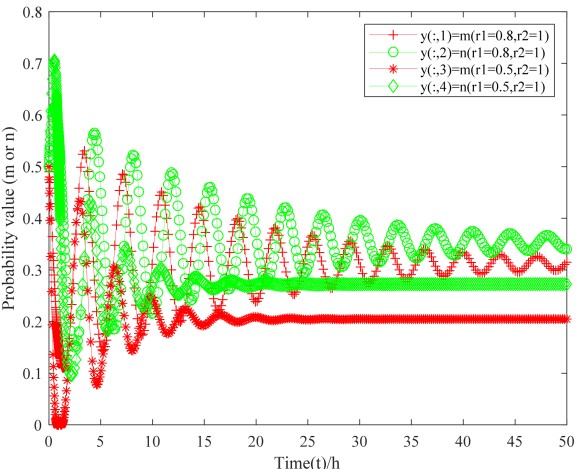

**Figure 4.** Decision-making simulation of positive insiders.

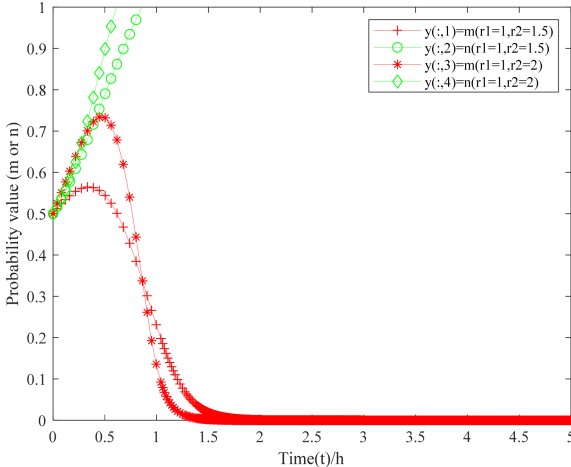

**Figure 5.** Decision-making simulation of pessimistic nuclear security department.

As shown in Figure 6, regardless of the emotional intensity of the nuclear security department, the evolution path will stabilize at a certain probability after the initial fluctuation. When $r_2 = 0.8$, the probability of insiders adopting the malicious behavior strategy is about 0.43, and the probability of strict implementation by the nuclear security department is about 0.27; that is, if the nuclear security department holds a slight optimism, the insiders are more inclined not to implement the malicious behavior strategy. However, when the emotional intensity of the nuclear security department increases (the emotional index $r_2$ decreasing), the probability of malicious behavior by insiders increases to about 0.5, and the probability of strict implementation by the nuclear security department decreases to about 0.06. The results show that the more optimistic the nuclear security department is about the situation of the internal insider's crime, the more inclined it is to believe that the internal insider will not betray the organization, thus weakening the vigilance against insiders and adopting the strategy of lax implementation. When rational insiders are aware of this, they will adopt a greater probability of malicious behavior and finally get closer to the worst strategy combination of the nuclear security department, forming a situation in which insiders collude to adopt the strategy of malicious behavior, but the nuclear security department does not strictly implement preventive and protective measures.

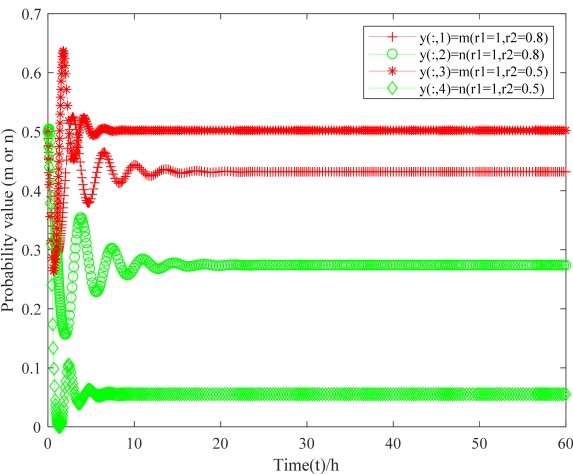

**Figure 6.** Decision-making simulation of positive nuclear security department.

*4.4. Insiders and the Nuclear Security Department Are in an Emotional State*

As shown in Figure 7, when $r_1 = 1.2$, $r_2 = 1.5$, that is, insiders and the nuclear security department were slightly pessimistic, the strict implementation probability of the nuclear security department quickly approaches one, while the probability of malicious behavior by the internal insider rapidly decreases to zero after a short increase. At this time, there is a stable strategy (strict implementation and not implementing malicious behaviors). Keeping the pessimistic intensity of the nuclear security department unchanged, the pessimistic intensity of the insiders was increased to 1.5 and 2 respectively, with different results. The two sides of the game finally realized the strategy combination (strict implementation and implementing malicious behaviors). This shows that the insiders with less pessimism are not completely irrational. When facing the nuclear security department with strong pessimism, the insiders believe that the nuclear security department will strictly implement prevention and protection measures and remain vigilant at all times, which will restrict their own behavior. However, when the pessimism of insiders becomes higher and higher, and they are aware of the strict implementation of the nuclear security department, they will have a stronger resistance and will adopt the strategy of malicious behaviors. Therefore, it is necessary to strengthen the emotional supervision of insiders as well as accurately study and judge the emotional state of insiders in order to make the best nuclear security decision.

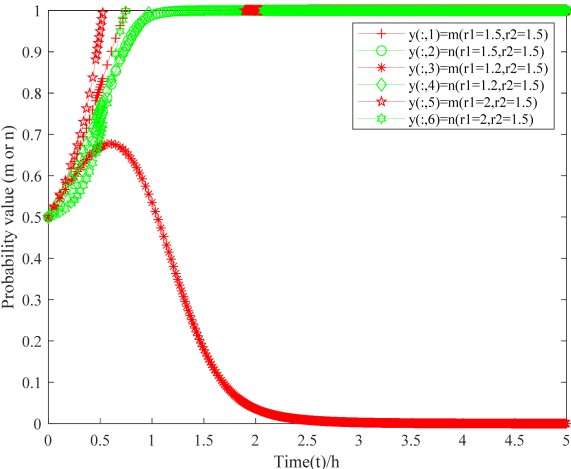

**Figure 7.** Decision-making simulation of pessimistic participants.

As shown in Figure 8, no matter what the optimism intensity of insiders and the nuclear security department is, it will not reach the evolutionary stable state. In addition, when $r_1 = 0.8$ and $r_2 = 0.8$, the probability of insiders adopting and implementing mali-

cious behaviors is about 0.37, and the probability of the nuclear security department strictly implementing preventive and protective measures is about 0.22. Keeping the emotional intensity of insiders unchanged at 0.8, when the optimistic emotional intensity of the nuclear security department is increased to 0.5 (the emotional index $r_2$ is decreasing), the probability of insiders implementing malicious behavior strategies is increased to about 0.45, and the probability of the nuclear security department strictly implementing prevention and protection measures is reduced to about 0.05. This means that, unlike the optimism of insiders, the nuclear security department remains rational. At this time, the strict implementation probability of the nuclear security department is lower than the malicious behavior probability of insiders; that is, the vigilance of the nuclear security department is insufficient, but the result is not stable. When the emotional intensity of the nuclear security department remains unchanged at 0.8 and the optimistic emotional intensity of insiders is increased to 0.5 (emotional index $r_1$ is decreasing), the probability of insiders implementing malicious behaviors is reduced to about 0.26, and the probability of strict implementation by the nuclear security department is reduced to about 0.16; that is, both parties are more inclined to choose (non-strict implementation and not implementing malicious behaviors), which is different from the above results. As insiders will experience an emotional change stage before committing malicious acts, the nuclear security department needs to be alert to the emotional change of insiders and appropriately improve the probability of strict implementation.

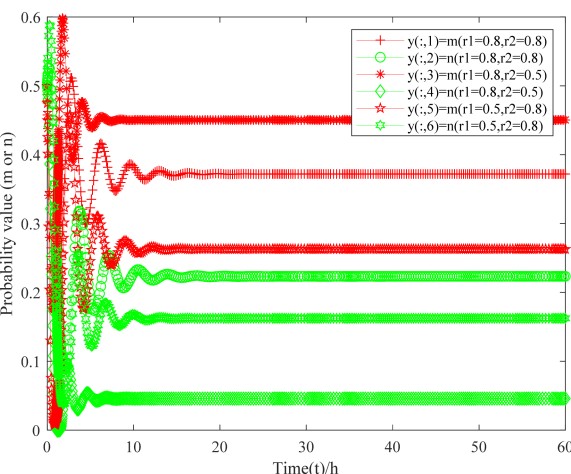

**Figure 8.** Decision-making simulation of positive participants.

## 5. Conclusions

The choices of insiders and the nuclear security department were affected not only by their own emotions, but also by each other's emotions, so there were different evolutionary equilibrium results. The optimism of the insider and the pessimism of the nuclear security department promote the evolution of the event to the result conducive to the nuclear security department, but the pessimism of the internal insider and the optimism of the nuclear security department may not lead to a bad result, which is related to the emotional intensity of the participants. Furthermore, the emotions of both sides will accelerate (pessimistic) or slow down (optimistic) the evolutionary game. Under the pessimistic mood, the emotional intensity of the nuclear security department will have a greater impact on the game equilibrium results. Therefore, it is very important to maintain a rational nuclear security department to prevent and protect internal threats. In addition, the nuclear security department needs to grasp the psychological state of the internal insider as accurately as possible, so as to make favorable nuclear security decisions. Therefore, strengthening emotional supervision and counseling is key in internal threat nuclear security.

This paper tentatively considers the impact of emotions of the game participants on the internal threat nuclear security events; however, there are deficiencies. Only the two-party

game between the internal insider and the nuclear security department was considered. In follow-up studies, we can consider adding third-party and the fourth-party actors to build a multi-agent model. Finally, the research on the internal factors of insiders needs to further consider the impact of non-emotional factors, such as the income level, moral quality and cultural level of participants on internal threat nuclear security events.

**Author Contributions:** Conceptualization, S.N.; Data curation, J.C.; Formal analysis, S.N. and S.Z.; Methodology, S.Z.; Software, J.C.; Supervision, S.Z.; Validation, S.Z.; Writing—original draft, S.N.; Writing—review and editing, S.N. All authors have read and agreed to the published version of the manuscript.

**Funding:** This research was funded by the Postgraduate Scientific Research Innovation Project of Hunan Province, China (grant No. CX20210946), the Research Foundation of Education Bureau of Hunan Province, China (grant No. 19A443) and Hunan Philosophy and Social Science Foundation Project (grant No. 14JD51).

**Institutional Review Board Statement:** Ethical review and approval were waived for this study due to no such events occurred.

**Informed Consent Statement:** Not applicable.

**Data Availability Statement:** Data are contained within the article.

**Acknowledgments:** The authors are grateful to Shoulong Xu for his guidance on the research. We also thank all the interviewees in the field visits.

**Conflicts of Interest:** The authors declare no conflict of interest.

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
