# Peer review of "Evolutionary Game Model of Internal Threats to Nuclear Security in Spent Fuel Reprocessing Plants Based on RDEU Theory"

_sustainability, doi:10.3390/su14042163_

Round 1

Reviewer 1 Report

Unfortunately, this paper's game theoretic model is not well-written. If the authors want to advance with this project, it is suggested that they consult a professional game theorist.  

Author Response

We rewrote the paper according to the editorial opinions, reorganized the logical framework of the paper, modified the grammatical errors, and made the article more coherent and complete.   

Thanks for your helpful suggestions.

Reviewer 2 Report

Generally speaking, this paper is well written and the analyses are thoroughly done. There are some questions that I want to ask the authors.

  1. I do not understand why WB(n) is included in (7) and (8). If it indicates the probability that player B uses strict implementation, it is strange that it does not coincide the decision weight shown in Table 5. The same thing applies in Eqs. (10) and (11). I think my understanding is not enough, but could you explain more? There seems to be not enough explanation in the text.
  2. I have an impression that the model in the paper can be used in any kind of regulation of illegal behaviors, not limited to nuclear security matters. In other words, the relationship between the contents of this paper and nuclear security matters is not so clear. I would like the authors to explain more about the reason why the paper focuses only on one specific matter.

I have some minor suggestions. "Hypothesis" in Section 3.1 could be change to "Assumption." In Table 2, σPND2S should be corrected to PND2S. Also, there are too much redundancy in writing of the paper, so the authors should consider to edit and revise it to reduce redundancy.

Reviewer 3 Report

The present manuscript describes, via the instrument of the Evolutionary Game,  the Internal Threats to Nuclear Security in 2 Spent Fuel Reprocessing Plant Based on RDEU Theory.

  1. Abstract: the reviewer suggets writting again this part of the manuscript including more literature background about the research topic of the manuscript.
  2. Moderate English changes are required.
  3. The research design, questions, hypotheses and methods should clearly stated
  4. The conclusiuons should be write again as a paragraph avoiding sub-sections and lists.

Round 2

Reviewer 2 Report

I am satisfied with the authors' responses and revisions.

Reviewer 3 Report

The present manuscript is well-written and fits well the aim and the scope of the journal

  • English language and style are fine/minor spell check required
  • Table layout should be re-thinking